# Coulomb Force from Non-Local Self-Assembly of Multi-Peak Densities in a Charged Space Continuum

Igor É. Bulyzhenkov [1,2]

1   Moscow Institute of Physics and Technology, National Research University, 141700 Dolgoprudny, Russia; ibphys@gmail.com
2   Physics Institute RAS, Peoples' Friendship University of Russia, 117198 Moscow, Russia

**Abstract:** Maxwell's electrodynamics admits radial charge densities of the elementary organization with one vertex of the spherical symmetry. A multi-vertex distribution of sharply inhomogeneous charge densities can also be described by monistic field solutions to Maxwell's equations–equalities. Coulomb–Lorentz forces are exerted locally to correlated electric densities in their volume organization with the fixed self-energy integral. The long-range Coulomb interaction between the dense peaks of the charged space continuum can be described quantitatively through bulk integrals of local tensions within observable bodies in favor of the monistic all-unity in the material space physics of Descartes and Russian cosmists.

**Keywords:** self-assembling; continuous charge; non-locality; local stresses; material space; monistic worldview





## 1. Introduction

Quantum mechanics works successfully with a correlated distribution of non-local matter in monistic terms of continuous wave functions. The local Schrödinger equation for the space–time dynamics of distributed matter is always supported by the integral conservation of continuous masses and charges in the entire spatial volume of the infinite Universe. This is the well-known normalization of the wave function for elementary matter and ensembles. Although observable bodies always consist of monistic distributions or overlapping quantum fields, the standard model of post-Newtonian physics retains the notion of localized charges and uncharged fields in order to interpret Maxwell's electrodynamic in the dual terms of Newtonian worldview.

In contrast to the dual physics of localized particles and massless/chargeless fields for distant interactions in empty space, the Cartesian metaphysics [1] and the monistic all-unity in Russian Cosmism [2–4] rely on the mobile ether between visible densities of material space. Needless to say, the concept of continuous matter–space of Russian cosmists (Lomonosov employed the gravitational liquid, Umov - the mobile ether and heat inertia, Tsiolkovsky - the monistic citizens of the entire Universe, Vernadsky - the Noosphere of material thoughts, etc.) is contrary to the Newtonian mainstream in academic textbooks on gravity and electrodynamics. Instead of the dual simplification through particles and fields, the monistic field alternative can describe any mass-energy integral $mc^2 = const$ by a continuous density $mc^2 n(\mathbf{x}, t, r_m = Gm/c^2) > 0$, which can acquire spherical symmetry, $mc^2 r_m/4\pi r^2(r + r_m)^2$, in time-averaged self-assembly or static equilibrium with one center of symmetry [5]. In a similar way, the nonlocal charge integral $q = const$ must also be distributed with a continuous density, $\int qn(\mathbf{x}, t, r_q)dV = q$, in all field points of the monistic matter–space. Such a material field physics, where the electric charge density $\rho \equiv qn(\mathbf{x}, t, r_q)$ and its current density are locally (i.e., without retardation) related to their electromagnetic fields $\mathbf{E}(\mathbf{x}, t, r_q)$ and $\mathbf{B}(\mathbf{x}, t, r_q)$ for all space–time coordinates, can transform Maxwell's equations for moving charges and time-varying chargeless fields into the local field equalities. The latter monistically describe charged densities by local electric

fields, $\rho(\mathbf{x}, t, r_q) \equiv div\mathbf{E}(\mathbf{x}, t, r_q)/4\pi$ but do not relate non-material fields with far-fetched distant particles.

One can recall that the classical charge density can be formally approximated by the delta operator $q\delta(\mathbf{x} - \boldsymbol{\xi}_q[t])$ for the assumed line path $\boldsymbol{\xi}_q[t]$ of the conventional point charge in the dual model of physical reality. This dual approach separates charges and their fields by different points in space. Such a practice-based separation of different bodies (or charged corpuscles) transforms the monistic unity of field equalities into the equations for coupled particles and fields, which become different concepts. The formal localization of charges seems to be suitable for approximating their dynamics in external fields, but it does not allow one to study the self-stresses of an isolated charge due to its own fields. In relativistic mechanics, the monistic theory for isolated mass-energy operates with the metric inertia for continuous densities of the Euclidean matter–space [6] but not with the metric gravitation of localized particles in the curved three-space. What contribution can the electric energy density make to the local stresses of a continuous space–matter with fixed integrals of mass and charge? The ethereal ideas of Plato, Aristotle, Descartes, Lomonosov, Umov, Tesla, Tsiolkovsky, Vernadsky and other cosmists have scientific rights to describe observations in monistic terms, as is customary for the quantum microcosm. Or should the standard model of macrophysics serve only Newton's dual worldview?

In this paper, we revisit the origin of the Coulomb force in non-local organizations of electrical energy and reveal the direct analogy with the local push by the Lomonosov gravitational liquid for the inverse square law of interaction. Again, we consider only the static moment in the non-equilibrium distribution of electric charge densities in their own fields and neglect the self-organization of continuous mass–energy densities of the small mechanical integral $mc^2$ next to the electrostatic self-energy when $(c^2/\sqrt{G})\sqrt{Gm^2} \ll (c^2/\sqrt{G})\sqrt{q^2}$.

In the monistic reading of Maxwell's equations–equalities for locally superimposed charge densities and their electric fields on macroscopic scales, each isolated charge element $q_k \equiv \int\rho(\mathbf{x}', t', q_k)d^3x'$ can have the equilibrium radial density $div\mathbf{E}/4\pi \equiv \rho \Rightarrow q_k r_k/4\pi r'^2(r' + r_k)^2 = E^2(r', r_k)/4\pi\varphi_o sign\, q_k$ in the co-moving coordinate system $\{\mathbf{x}', t'\}$. Hereinafter, $\varphi_o \equiv c^2/\sqrt{G}$ is the universal self-potential for mechanical and electric charges, while $r_k \equiv \sqrt{(q_k^2 + Gm_k^2)}/\varphi_o$ ($\approx |q_k|\sqrt{G}/c^2$ for $\sqrt{G}m_k \ll |q_k|$) is the spatial scale for these charges. Elementary charge organizations can have different magnetic moments, which is possible from charged densities with toroidal rotations on microscales. In the monistic field approach to electricity, we will be primarily interested in understanding the origin of long-range interactions between static microscopic charges. For the sake of simplicity, all dynamical options in macroscopic interactions will be neglected for the time being.

## 2. Monistic Continuum Method with Multiple Density Peaks

To understand the all-unity of the world system and its self-governance principles, it is useful to analyze the non-local repulsion of two continuous charges $q_1 \equiv \varphi_o r_1 = \int\rho_1(\mathbf{x}, 0)d^3x > 0$ and $q_2 \equiv \varphi_o r_2 = \int\rho_2(\mathbf{x}, h)d^3x > 0$ (Figure 1) within their isolated (closed) system. We consider only a moment of time for the static distribution of two elementary densities in one system in order to calculate the Coulomb counter-actions without contributions of magnetic forces.

The post-Coulomb potential of strong electric fields in a two-peak distribution of the system charge integral can be modeled,

$$W_{sys}(\mathbf{r}, \mathbf{h}) = \varphi_o \ln\left(1 + \frac{r_1}{|\mathbf{r}|} + \frac{r_2}{|\mathbf{r} - \mathbf{h}|}\right), \tag{1}$$

from the static field analogy to the complementary system of continuous masses. There, the metric potential of the static continuum $W_{met} = -\varphi_o ln(1 + \sum \sqrt{G}m_k/|\mathbf{x} - \mathbf{a}_k|\varphi_o)$ was already derived for the pseudo-Riemann matter–space–time of continuous mass–energy [5].

The logarithmic potential (1) for a multi-peak distribution of the system charge densities corresponds to asymmetric electric fields,

$$\mathbf{E}_{sys}(\mathbf{h}) \equiv -\nabla W_{sys}(\mathbf{h}) = \frac{\varphi_o}{1 + \frac{r_1}{|\mathbf{r}|} + \frac{r_2}{|\mathbf{r}-\mathbf{h}|}} \left( \frac{\mathbf{r}r_1}{|\mathbf{r}|^3} + \frac{(\mathbf{r}-\mathbf{h})r_2}{|\mathbf{r}-\mathbf{h}|^3} \right) \equiv \mathbf{E}_1 + \mathbf{E}_2, \qquad (2)$$

in all points of the charged 3-space $\mathbf{x} \equiv \mathbf{r}$. By dividing the static strength (2) into two 'elementary' fields, $\mathbf{E}_{sys} = \mathbf{E}_1 + \mathbf{E}_2$, one can formally divide the 'whole-one all-unity' into two 'elementary' densities, $\rho_{sys} \equiv div\mathbf{E}_{sys}/4\pi = (div\mathbf{E}_1/4\pi) + (div\mathbf{E}_2/4\pi) \equiv \rho_1 + \rho_2$. From such definitions, we can find continuous densities of the non-local energy unity $(q_1 + q_2)\varphi_o > 0$,

$$\rho_{q_{sys}} \equiv \frac{\mathbf{E}_1^2}{4\pi\varphi_o} + \frac{\mathbf{E}_1\mathbf{E}_2}{2\pi\varphi_o} + \frac{\mathbf{E}_2^2}{4\pi\varphi_o} = \frac{\varphi_o\left( \frac{r_1^2}{|\mathbf{r}|^4} + \frac{2r_1 r_2 \mathbf{r}(\mathbf{r}-\mathbf{h})}{|\mathbf{r}|^3|\mathbf{r}-\mathbf{h}|^3} + \frac{r_2^2}{|\mathbf{r}-\mathbf{h}|^4} \right)}{4\pi\left(1 + \frac{r_1}{|\mathbf{r}|} + \frac{r_2}{|\mathbf{r}-\mathbf{h}|}\right)^2} \geq 0, \qquad (3)$$

and formal 'elementary' densities of two contributing elements $q_1$ and $q_2$,

$$\rho_{q_1} \equiv \frac{\mathbf{E}_1^2 + \mathbf{E}_1\mathbf{E}_2}{4\pi\varphi_o} = q_1\left( \frac{r_1}{|\mathbf{r}|^4} + \frac{r_2\mathbf{r}(\mathbf{r}-\mathbf{h})}{|\mathbf{r}|^3|\mathbf{r}-\mathbf{h}|^3} \right) \bigg/ 4\pi\left(1 + \frac{r_1}{|\mathbf{r}|} + \frac{r_2}{|\mathbf{r}-\mathbf{h}|}\right)^2, \qquad (4)$$

$$\rho_{q_2} \equiv \frac{\mathbf{E}_2^2 + \mathbf{E}_1\mathbf{E}_2}{4\pi\varphi_o} = q_2\left( \frac{r_2}{|\mathbf{r}|^4} + \frac{r_1\mathbf{r}(\mathbf{r}-\mathbf{h})}{|\mathbf{r}|^3|\mathbf{r}-\mathbf{h}|^3} \right) \bigg/ 4\pi\left(1 + \frac{r_1}{|\mathbf{r}|} + \frac{r_2}{|\mathbf{r}-\mathbf{h}|}\right)^2. \qquad (5)$$

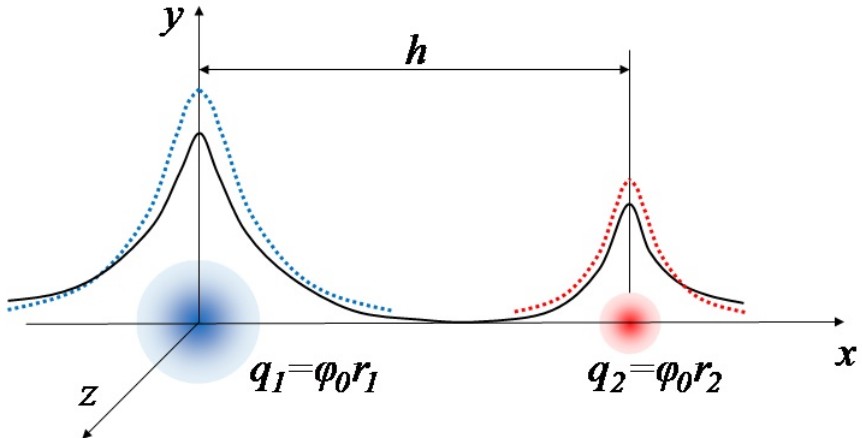

**Figure 1.** The dotted lines are equilibrium radial densities for $h \to \infty$; the solid curve is for the nonequilibrium density of non-local charge–energy with asymmetric tensions.

The continuous distribution (3) for the non-local self-organization of the system energy $(q_1 + q_2)\varphi_o$ does not have negative (non-physical) densities, $\rho_{sys} \geq 0$ (with only one zero density point at $\mathbf{E}_1 = -\mathbf{E}_2$). An example of everywhere positive densities in (3) for $q_1/q_2 = r_1/r_2 = 5$ and $|\mathbf{h}| = 2r_1$ is shown in part A of Figure 2. Both 'elementary' densities (4) and (5) in the formal partition $\rho_{sys} = \rho_1 + \rho_2$ have regions with negative charge densities (part B in Figure 2) and, therefore, with negative 'elementary' densities $\rho_k\varphi_o$ of the self-energy integral $q_k\varphi_o > 0$. The point is that the 'elementary' densities (4) and (5) do not represent isolated or elementary objects within the non-local all-unity. As in Eastern Holism, the monistic all-unity in Russian Cosmism cannot be separated into completely independent parts of the united world energy. The density (4), for example, contains the one-vertex (positive) and two-vertex (interference, shared) fractions. The interference fraction of 'elementary' energy densities admits negative values (for $0 < x < h$) which underline non-equilibrium tension areas in the charged space.

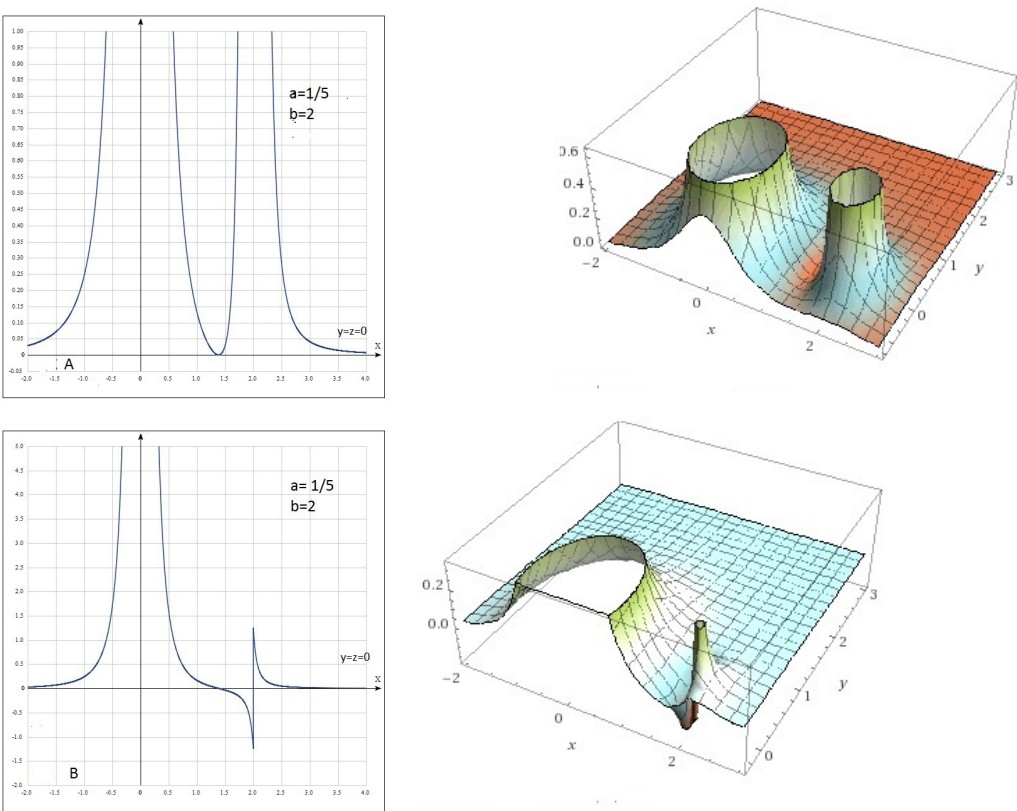

**Figure 2.** $a \equiv r_2/r_1$, $b \equiv h/r_1$. **(A)** The self-governed density (3) is never negative, $\rho_{sys}(x,y,0) / \frac{q_1}{4\pi r_1^3}$

$$= \left( \frac{1}{(x^2+y^2)^2} + \frac{2a[x(x-b)+y^2]}{(x^2+y^2)^{3/2}[(x-b)^2+y^2]^{3/2}} + \frac{a^2}{[(x-b)^2+y^2]^2} \right) / \left( 1 + \frac{1}{\sqrt{x^2+y^2}} + \frac{a}{\sqrt{(x-b)^2+y^2}} \right)^2 \geq 0.$$

**(B)** $\rho_1(x,y,0) / \frac{q_1}{4\pi r_1^3} = \left( \frac{1}{(x^2+y^2)^2} + \frac{a[x(x-b)+y^2]}{(x^2+y^2)^{3/2}[(x-b)^2+y^2]^{3/2}} \right) / \left( 1 + \frac{1}{\sqrt{x^2+y^2}} + \frac{a}{\sqrt{(x-b)^2+y^2}} \right)^2.$

There is no doubt that the charge integral $\int \rho_{sys} d^3 x = q_1 + q_2 = const$ and the electric self-energy $(q_1 + q_2)\varphi_o = const$ of the closed system do not depend on the mutual distance $h$ between the centers of elementary distributions in Figure 1. However, what about the possible constancy of each 'elementary' continuous charge $q_k$ in paired Coulomb interactions as is customary for the classical theory of point charges? The finite vector $\mathbf{h} \equiv \hat{\mathbf{h}}h$ is responsible for elastic changes in the spatial asymmetry of initially radial fields $\mathbf{E}_k^\infty = \hat{\mathbf{r}}q_k/r^2[1 + (q_k/r\varphi_o)]$ of an isolated (equilibrium) charge at $h \to \infty$. Spatial integration into one non-local system makes 'elementary' distributions (4) and (5) asymmetric depending on the direction and magnitude of $\mathbf{h}$. This non-equilibrium asymmetry of 'elementary' charges, concentrated around individual peaks of the continuous distribution of the integral-conserved charge of the entire system, leads to a self-consistent rearrangement of all correlated densities when their observed peaks move, which in practice is perceived as a Coulomb repulsion of visible bodies along a contrived long-range action. In the next section, we calculate these Coulomb forces from the local self-tensions of the continuous charged space of the non-local distribution with integral conservation of the system energy. This way, the classical action-at-a-distance of continuous charges–extensions will come from the non-local self-organization of their adaptive self-energies in the indivisible all-unity.

First, we analyze new topology properties of 'elementary' energies in their multi-vertex system. The energy density $\rho_1\varphi_o$, for example, contains in (4) two topologically different contributions, namely one-vertex density $\mathbf{E}_1^2/4\pi$ and two-vertex density $\mathbf{E}_1\mathbf{E}_2/4\pi$. The constant 'elementary' charge $q_k$ (or its electric self-energy $q_k\varphi_o$) gains two substantive parts exclusively due to its involvement into the multi-vertex system. Despite the integral energy conservation, any system is not a sum of isolated (radial) elements. The latter interact and acquire interference energy terms $\mathbf{E}_1\mathbf{E}_2/4\pi$ with a two-vertex topology. Approaching

a non-local system from infinity, each initially radial element contributes to the interference energy of the shared material space from its self-energy integral $q_k \varphi_o = const$. Now, we check the elementary energy conservation by calculating from (4) its one-vertex and two-vertex topological parts at various proximity h:

$$\begin{cases} \int \frac{d^3x}{4\pi} \mathbf{E}_1^2 + \int \frac{d^3x}{4\pi} \mathbf{E}_1 \mathbf{E}_2 = \varphi_o^2 r_1 \left(1 - \frac{r_2}{h}\right) + \frac{\varphi_o^2 r_1 r_2}{h} = q_1 \varphi_o = const, \ r_1, r_2 \ll h \to \infty, \\ \\ \int \frac{d^3x}{4\pi} \mathbf{E}_1^2 + \int \frac{d^3x}{4\pi} \mathbf{E}_1 \mathbf{E}_2 = \frac{\varphi_o^2 r_1^2}{r_1 + r_2} + \frac{\varphi_o^2 r_1 r_2}{r_1 + r_2} = q_1 \varphi_o = const, \ r_1, r_2 \gg h \to 0. \end{cases} \tag{6}$$

These integrals of 'elementary' self-energies are constant for both calculated limits in (6). Such a mathematical result corresponds to the expected conservation $q_k = const$ of each continuous sub-charge in elastic interactions. Certainly, there is self-energy conservation, $\sum q_k \varphi_o = const$, of the closed system in spite of mutual interactions of its 'elements'. Conceptually, there are no separated elements or particles–corpuscles in the monistic all-unity of the world material field in Russian Cosmism. Elementary corpuscles with the loss of system properties appear after the process of isolating these corpuscles (say, during local macro-measurements in the laboratory) as new systems. The cosmic hierarchy of isolated systems in the non-local Universe can be formally classified through the integral conservation of volume self-energy for the correlated self-organization of material densities. Each system cannot in reality be completely holonomic, which corresponds to Gödel's incompleteness theorem.

### 3. Local Self-Tensions of a Continuously Charged Space Result in the Volumetric Forces of Coulomb's Law

According to the non-empty space doctrine of Russian Cosmism, there is no uncharged space for the charged all-unity with one, two or more peaks (vertexes) of continuous densities. In this monistic physics of purely field space–matter, one can talk about the local self-tension ($\rho_{sys} \mathbf{E}_{sys}$ in statics) within the non-local continuum rather than about the Lorentz force for the charge density in external fields. The correlated self-organization of monistic fields and currents at all space–time points preserves the volume integrals of self-energies or electric charges in holonomic systems. The question at hand is how to calculate the volume force ($\int_\Omega \rho_{sys} \mathbf{E}_{sys} d^3x$) for a given volume $\Omega$ of continuously charged densities. This integral force can be estimated for extremely dense, visible volumes of charged space–matter. Two such volumes should be enough to calculate the Coulomb force analog in the monistic field approach. The most concentrated densities of the system in Figure 1 are around the origin $\{0,0,0\}$ and around the second vertex $\{h,0,0\}$. Recall that both integrals $q_1 \varphi_o$ and $q_2 \varphi_o$ are continuously distributed self-energies over the infinite charged space of our two-body system.

Now, let us calculate the electric force integral exerted along the x-direction to the densest sphere $\Omega_1$ around the origin. For large (macroscopic) distances h, when $|q_1| / \varphi_o \ll \Omega_1^{1/3} \ll h$, one can find the following repulsion of superimposed positive charges:

$$F_{\Omega_1}^x = \int_{\Omega_1} \rho_{sys} E_{sys}^x dxdydz = \int_{\Omega_1} \frac{dxdydz}{4\pi\varphi_o} (\mathbf{E}_1^2 + 2\mathbf{E}_1\mathbf{E}_2 + \mathbf{E}_2^2)(E_1^x + E_2^x) \approx$$

$$\int_{\Omega_1} \frac{dxdydz}{4\pi\varphi_o} (\mathbf{E}_1^2 E_1^x + 2\mathbf{E}_1\mathbf{E}_2 E_1^x + \mathbf{E}_1^2 E_2^x) = -\frac{\varphi_o^2 r_1 r_2}{h^2} \left(\frac{1}{6} + \frac{1}{3} + \frac{1}{2}\right) = -\frac{q_1 q_2}{h^2}. \tag{7}$$

Here, we omit the vector tensions $\mathbf{E}_2^2 E_1^x / 4\pi\varphi_o$ and $(2\mathbf{E}_1\mathbf{E}_2 + \mathbf{E}_2^2) E_2^x / 4\pi\varphi_o$ because of their insignificant integrals over the ultra small volume $\Omega_1 = 4\pi(100r_1)^3/3$ around the origin. The measurable force (7) for the volume $\Omega_1$ originates from the local self-action of the non-local distribution in the very same volume $\Omega_1$ with the peak densities at $\{0,0,0\}$. Half of this volumetric force gives the field density $\mathbf{E}_1^2 E_2^x / 4\pi\varphi_o$ with the intensity $E_2^x$ of the second charge. The other half comes from the asymmetric self-action $\mathbf{E}_1^2 \mathbf{E}_1 / 4\pi\varphi_o$ and the local interference $\mathbf{E}_1\mathbf{E}_2 E_1^x / 2\pi\varphi_o$ in self-pushing material space.

Now, consider the bulk stresses in the ultra-small volume $\Omega_2 = 4\pi(100r_2)^3/3$ around the vertex $\{h, 0, 0\}$ in order to analytically calculate the corresponding force at $h \gg 100r_2$:

$$
\begin{aligned}
F_{\Omega_2}^x &= \int_{\Omega_2} \rho_{sys} E_{sys}^x dxdydz = \int_{\Omega_2} \frac{dxdydz}{4\pi\varphi_o} (\mathbf{E}_1^2 + 2\mathbf{E}_1\mathbf{E}_2 + \mathbf{E}_2^2)(E_1^x + E_2^x) \\
&\approx \int_{\Omega_2} \frac{dxdydz}{4\pi\varphi_o} (\mathbf{E}_2^2 E_2^x + 2\mathbf{E}_1\mathbf{E}_2 E_2^x + \mathbf{E}_2^2 E_1^x) = \frac{\varphi_o^2 r_1 r_2}{h^2}\left(\frac{1}{6} + \frac{1}{3} + \frac{1}{2}\right) = \frac{q_1 q_2}{h^2}.
\end{aligned}
\tag{8}
$$

This measurable force also contains the local self-action of the strong asymmetrical field $\mathbf{E}_2$ and local self-pushes in a non-locally organized distribution of continuous energy. The local self-tension $\rho_{sys}E_{sys}^{y,z}$ is antisymmetric for $y \to -y$ or $z \to -z$, which nullifies the integral forces orthogonal to the axis $\hat{\mathbf{h}}$, $F_{\Omega_1}^{y,z} = 0$ and $F_{\Omega_2}^{y,z} = 0$.

In macroscopic practice, there are no easy ways to distinguish between ethereal self-actions (7)–(8) in monistic physics and remote interactions through the material void in dual physics. So far, the monistic unity of the kinetic ether in Russian Cosmism with local Lomonosov pushes does not withstand computational simplifications from the dualistic approach to material particles and chargeless/massless fields-forces in the void. Since their time in high school, many physicists sincerely believe that the dualistic Standard Model of particles and field mediators is reliably confirmed by measurements. In fact, any experiment can only falsify a theory, but never confirm it, as Karl Poppers and other thinkers have repeatedly argued. Laboratory measurements make it possible to distinguish between Newton's empty space with "action-at-a-distance" and Umov's energy continuum with local impacts of Lomonosov's liquid-matter by refuting quantitatively one of the competing approaches beyond a reasonable doubt.

## 4. Discussion

Shannon's theory determined the optimal distribution of information amount $P$ over noise lines $wN_o$ in the same logarithmic form, $C(N_o) = wlog_2[1 + (P/wN_o)]$, as our potential tension (1), $W(r) = \varphi_o ln[1 + (q/\varphi_o r)]$. This information law determines the equilibrium distribution of energy densities $\varphi_o\rho(r) = \varphi_o\nabla^2 W(r)/4\pi = q^2\varphi_o^2/4\pi r^2(r\varphi_o + q)^2$ over the field continuum of electric self-energy $q\varphi_o$ with the embedded information. To trace the analogy of information and energy distributions around one center of spherical symmetry, it suffices to assume that the noise cycle $N_o$ is proportional to radial distance $r$ from this center to the periphery. The corresponding bandwidth $\varphi_o$ for energy communications inside the self-assembling shape can be defined as the self-potential $c^2/\sqrt{G}$ of the non-local electric charge.

The logarithmic saturation of information channels can be verified in practice due to the technical possibility of changing the transmission width $w$ in modern telecommunication systems. The world organization of continuous mechanical and electrical energies has fixed the transmission potential by one universal value $\varphi_o \equiv c^2/\sqrt{G}$. This extremely broad bandwidth provides a harmonic approximation of the Shannon potential (1) for almost all observations, which mimics empty space physics for weak (measurable) fields with $\nabla^2 W(\mathbf{x}) \approx 0$. However, the mathematical analogy between information and energy potentials can reveal the continuous distribution of Shannon information in the material medium and offer new laboratory tests of eddy matter-space with adaptive responses of correlated densities in the monistic theory of metric inertia [6].

From Shannon's information theory, we can conclude that the weaker the signal-to-noise ratio $\sum_k q_k \sqrt{G}/\varphi_o|\mathbf{r} - \mathbf{a}_k|$, the lower the energy exchange capacity in the charged matter–space of Russian Cosmism. Therefore, the energy–information continuum should not be very sparse for reliable transmission of significant amounts of electrical/mechanical power and Shannon information. It was Plato who first denied emptiness, saying in Timaeus that "matter and space are the same". Subsequently, Aristotle logically concluded that the movement of material bodies from one place to another cannot pass through voids or empty space. Same as in the teaching of ancient Greeks, there is no empty space in the monistic physics of Russian Cosmism. And, consequently, there is no energy-information

transmission through the void in the teachings of M. Lomonosov, N. Umov, K. Tsiolkovsky, D. Mendeleev, V. Vernadsky, A. Chizhevsky. Superpenetrating ether is associated with ponderable properties, local dynamics, nonlocal correlations and reversible thermomechanics even in the cosmic extension of the human superimposed on the Noosphere of material thoughts. Everyone was proclaimed an ethereal citizen of the entire Universe in the monistic all-unity of living and inert matter.

In the monistic charge-field approach to Maxwell's continuous electricity, the Coulomb 'action-at-a-distance' originates from local tensions of the non-local multi-vertex unity of continuous charge energy. The system self-energy $\sum q_k \varphi_o = q_{sys} \varphi_o$ is constant under all distances between vertexes of continuously distributed 'elements' $q_{sys} = const$. Different bulk regions of the non-local self-assembly gain different self-forces depending on correlated positions (and velocities, in general) of elementary vertexes in steady chemical compositions. The bulk force from non-equilibrium tensions of the densest region of the inhomogeneously charged space around each peak formation with a point vertex equally depends on the asymmetric distributions of its own electric field and external fields formally associated with distant vertexes.

The system self-governance of correlated tensions, bulk forces of selected volumes, and non-local redistribution of 'elementary' charge densities occurs under the wave-free conservation of elastic sub-energy $q_k \varphi_o = const$. The latter is an electric analog of the positive (kinetic) rest-energy $m_k c^2 = const$ of each extended mass in the non-local mechanical system. The spatial transport of charged densities is proportional to the transport of the corresponding electric self-energies with the universal potential $c^2 / \sqrt{G}$. The correlated asymmetry of the self-assembling densities ($E_1^2 E_1^x + 2E_1 E_2 E_1^x$) in the force integral (7) can be controlled by local resonances with external power interventions. Therefore, this half of the Coulomb 'action-at-a-distance' can be controlled in practice by local intervention in the asymmetry of energy densities, which is essential for long-range interactions in the monistic physics of Russian Cosmism. The mental control of this subtle asymmetry in Lomonsov's ethereal pushes (7)–(8) can give some theoretical insight into telekinesis and Kadochnikov's technique for non-contact hand-to-hand combat, patented in 2000.

Each isolated charge $q_k$ accepts a radial distribution of elementary densities from Maxwell's equations–equalities but asymmetric densities (4) or (5) in a two-vertex system at $h \neq \infty$. The stationary motion of continuous electrons must be accompanied by a rigid motion of their electric fields embedded in the geometric structure of charged densities in monistic physics. The rigidity of the Coulomb field was first discovered for an ultra-relativistic electron beam [7]. Such experiments can be modified [8] in order to falsify either the non-empty space of monistic physics or the empty-space of dualistic physics. In the macroscopic nonlocality of cosmic systems, it is important to distinguish dissipative exchanges of retarded/advanced waves from instantaneous correlations (7)–(8) of elastic fields. The ethereal physics of the non-local whole can reasonable describe all three signals (retarded, instantaneous, advanced) in Kozyrev's telescopic experiments [9].

The non-local self-assembly of continuously charged matter–space can be studied in gravitation-free cosmos for the visible (macroscopic) peaks of continuous energy profiles. Experimental studies of laboratory-controlled interactions in self-organized plasma in microgravity [10] can, in principle, shed new light on the kinetic mechanism of Lomonosov self-pushes in continuous space–matter. More generally, a continuously charged space with Coulomb interactions (7)–(8) due to instantly correlated stresses over the volume integral of non-local self-energy supports the monistic world reality for fundamental and applied problems in physics, chemistry, biology and other disciplines.

**Funding:** This research received no external funding.

**Data Availability Statement:** All data generated or analyzed during this theoretical study are included in this published article.

**Conflicts of Interest:** The author declares no conflict of interest.

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
