# Peer review of "Coulomb Force from Non-Local Self-Assembly of Multi-Peak Densities in a Charged Space Continuum"

_2571-712X, doi:10.3390/particles6010007_

Round 1

Reviewer 1 Report

I have read with interest this work, which was written in a fluent and understandable way. In this manuscript, the author focused his attention on revisiting the origin of the Coulomb force in nonlocal organizations of electrical energy and revealing a direct analogy via the local push by the Lomonosov gravitational liquid for the inverse square law of interaction. I also scanned the submission through the iThenticate system and found that there were no ethical issues. Since the results presented here are interesting, I think the study could be published in this journal. On the other hand, I would like to draw attention to one point. The reference in the last line of page 2 is missing and needs to be corrected.

Author Response

Thanks for your review. I fixed the issue and corrected the references.

Reviewer 2 Report

The thesis of the monistic physics of the field-source relation is interesting despite its apparence to be eccentric, anyay the author do not go in deep with the consequence regarding the Local Lorentz Invariance but only gives a glance to issues referring to causality mainly about the retarded potential.

The auto references despite their necessity could seem in excess but since this topic is so particular can be accepted.

The overall pity is the lack of a sound proposal of experimental test even at ideal (the so called gedanken) level.

In summary a nice physical-mathematical work well done and described, which deserves publication as a seminal work along the tradition of the Russian school of physics, which could get the attention of the readers of the review Particle mainly the experimentalists in particle physics who can start from the idea presented in this work to test the issue of retarded potential in  the fundamental interactions.

Author Response

Yes, I replaced some of my auto references with the references on other domesic cosmists (Lomonosov, Umov, Tsiolkovsky, Kozyrev).

I agree with the lack of direct experimental tests and put few words (in the  section  Discussion) where this monistic approach can shed some light on the selected phenomena in practice.